# Marker-Assisted Hybridization and Selection for Fiber Quality Improvement in Naturally Colored Cotton (*G. hirsutum* L.)

**DOI:** 10.3390/plants14233601

**Published:** 2025-11-26

**Authors:** Abrorjon Y. Kurbonov, Feruza F. Mamedova, Muxammad-Latif M. Nazirov, Naima Sh. Khojaqulova, Sanjar Sh. Djumaev, Nigora R. Khashimova, Barno B. Oripova, Asiya K. Safiullina, Ezozakhon F. Nematullaeva, Kuvandik K. Khalikov, Dilrabo K. Ernazarova, Fakhriddin N. Kushanov

**Affiliations:** 1Cotton Breeding, Seed Production and Agrotechnologies Research Institute, University Street, Kibray, Tashkent 111218, Uzbekistan; feruzamamedova2989@mail.ru (F.F.M.); nazirovlatif015@gmail.com (M.-L.M.N.); naimaxojaqulova90@mail.ru (N.S.K.);; 2Institute of Bioorganic Chemistry Named After O. Sodikov, Academy of Science of the Republic of Uzbekistan, Tashkent 100125, Uzbekistan; nigora.khashimova@gmail.com; 3Institute of Genetics and Plant Experimental Biology, Academy of Sciences of the Republic of Uzbekistan, Tashkent 111208, Uzbekistan; barnooripova127@gmail.com (B.B.O.); asiyasafiullina0996@gmail.com (A.K.S.); ezoza.fn@gmail.com (E.F.N.); quvondiqxaliqov87@gmail.com (K.K.K.); edilrabo64@gmail.com (D.K.E.); 4Department of Histology and Medical Biology, Tashkent State Medical University, Tashkent 100109, Uzbekistan; 5Department of Biotechnology and Microbiology, National University of Uzbekistan, Tashkent 100174, Uzbekistan; 6Department of Genetics, Samarkand State University Named After Sh. Rashidov, Samarkand 140104, Uzbekistan

**Keywords:** *G. hirsutum* L., naturally colored cotton, crossing, fiber quality, SSR marker, BNL1604, phenotypic evaluation, marker-assisted selection, genetic diversity, sustainable breeding

## Abstract

Naturally colored cotton offers ecological advantages by eliminating the need for chemical dyeing; however, its limited fiber quality restricts its commercial utilization. The main goal of this study was to evaluate the potential of the SSR marker BNL1604 for marker-assisted selection in naturally colored cotton (*G. hirsutum* L.) and to assess fiber quality variation among hybrid progenies derived from crosses between colored and elite white-fiber cultivars. As an expected outcome of this approach, we also assessed whether hybridization of naturally colored lines with elite white-fiber cultivars could contribute to the improvement of fiber quality traits in segregating progenies. Five colored lines (brown and green), three elite cultivars, and fifteen derived F_3_ progenies were analyzed. Fiber traits, including upper half mean length (UHML), strength, elongation, and micronaire, were measured using HVI. Genotyping was conducted with BNL1604, and in silico mapping localized this marker to chromosome A07, with a homoeologous region on D07. White-fiber cultivars exhibited superior fiber length (33.4–35.4 mm) and strength (>31 g·tex^−1^) compared with colored lines. Several F_3_ hybrids exhibited transgressive segregation (progeny with trait values significantly exceeding those of both parents, as confirmed by frequency distribution and ANOVA analyses). For instance, the F_3_ (C-6577 × L-4099) hybrid achieved UHML values of 30.51 mm and strength > 31.93 g·tex^−1^. Most progenies maintained optimal micronaire (4.0–4.9). It was concluded that the presence of the 107 bp allele of BNL1604 marker was strongly associated with high-quality fiber, specifically improved fiber strength and length. In silico annotation revealed candidate genes near the BNL1604 locus linked to fiber development. These findings highlight the potential of combining hybridization with selection based on the presence of this 107 bp allele to develop high-quality, naturally colored cotton cultivars.

## 1. Introduction

Cotton (*Gossypium* spp.) represents one of the most genetically diverse and economically important textile crops, serving as a key model for studies on genome evolution, interspecific hybridization, and fiber quality improvement [1,2,3]. In recent years, naturally colored cotton (NCC), encompassing brown, green, and other pigmented fiber types, has garnered considerable attention as an environmentally sustainable alternative to conventional white cotton. The ecological appeal of NCC lies in its ability to eliminate the use of synthetic dyes and chemical dyeing processes, thereby reducing environmental pollution [4]. The natural pigmentation predominantly results from the accumulation of metabolites, including flavonoids and proanthocyanidins, within the fiber, regulated by specific biosynthetic pathways [5,6,7,8].

Despite these advantages, NCC genotypes typically exhibit inferior fiber quality compared to white-fiber cotton (WFC) cultivars. NCC accessions generally suffer from poor fiber quality, including shorter fiber length, lower strength, and higher micronaire compared with WFC cultivars [9,10,11]. These limitations significantly hinder the commercial utilization of NCC in the production of high-grade textiles.

In Uzbekistan and other cotton-producing regions, breeding programs targeting naturally colored cotton date back to the 1950s, leading to the development of lines with diverse fiber pigmentation, including brown, reddish, and green shades [4,12,13,14]. However, the commercialization of these lines has been constrained by their inadequate fiber length and strength [11]. Consequently, enhancing the fiber quality of NCC while preserving its natural pigmentation remains both a significant breeding challenge and a priority objective for cotton improvement programs [4].

According to Wang et al. [15], hybridization between NCC lines and elite WFC cultivars represents an effective strategy for improving fiber quality. By crossing pigmented-fiber lines-harboring desirable pigmentation genes-with high-quality WFC cultivars, breeders aim to introgress favorable fiber traits such as increased length, enhanced strength, and improved fineness into the NCC genetic background [16]. Fiber quality parameters are typically quantitative, polygenic, and strongly influenced by environmental conditions [17,18]. Among these, upper half mean length (UHML), fiber strength (FS), fiber elongation (FE), and fiber micronaire (FM, an indicator of fiber fineness and maturity) are critical determinants of spinning efficiency and textile performance [19]. Given that WFC cultivars generally outperform pigmented lines in these traits, the introgression of favorable alleles into naturally colored cotton remains a fundamental and ongoing breeding objective [4].

In recent decades, advances in DNA marker technologies and molecular mapping have provided powerful tools to accelerate cotton breeding [20,21]. Numerous quantitative trait loci (QTLs) associated with fiber quality traits have been identified in diverse mapping populations [1,22]. Molecular linkage analysis and genome-wide association studies (GWAS) have mapped QTLs for fiber length, strength, and micronaire to multiple chromosomes of *G. hirsutum* [23,24]. However, many QTLs display genotype- or environment-specific effects, which restricts their utility in practical breeding applications. Only stable QTLs with large phenotypic effects are considered reliable targets for marker-assisted selection (MAS) [25,26,27,28].

Simple sequence repeat (SSR) markers remain among the most widely used DNA markers in cotton genetics and breeding due to their PCR-based detection, co-dominant inheritance, and high polymorphism, making them particularly suitable for genotyping the polyploid cotton genome [29,30]. Thousands of SSR markers have been developed, among which the Brookhaven National Laboratory (BNL) series, released in the early 2000s, has been extensively employed in cotton genetic mapping [23,24]. One marker of particular interest is BNL1604, which contains an AG dinucleotide repeat motif and is mapped to homeologous chromosomes A07 and D07 of allotetraploid cotton. BNL1604 has consistently shown linkage or association with key fiber traits-including fiber length (FL), fiber strength (FS), fiber uniformity (FU), and micronaire (FM)-across multiple studies [1,31]. For example, Tan et al. [23] and Fang et al. [24] reported QTLs for fiber length and strength linked to the BNL1604 locus, while Abdurakhmonov et al. [1,32] identified a ~101 bp allele associated with superior fiber length, strength, and fineness in exotic germplasm. Consequently, BNL1604 has been widely utilized in cotton MAS programs aimed at improving fiber quality. In Uzbekistan, recent breeding programs have successfully tracked favorable BNL1604 alleles through backcross generations to develop new *G. hirsutum* cultivars with improved fiber length and strength [1,26,32], further underscoring its relevance for cotton improvement.

Despite the availability of such valuable markers, their application in naturally colored cotton breeding remains limited. Conventional breeding methods continue to dominate, while molecular tools have not yet been widely implemented in colored cotton improvement. Given the generally low heritability of fiber traits in pigmentated backgrounds, integrating MAS could significantly enhance breeding efficiency. In this context, BNL1604 represents a promising marker for tracking favorable alleles in crosses between white and naturally colored cotton.

The present study aimed to evaluate fiber quality improvements within and among F_3_ hybrid progenies derived from crosses between NCC accessions and WFC cultivars of *G. hirsutum* L. Specifically, fiber length, strength, elongation, and fineness were evaluated, and the association of these traits with allelic variation at the BNL1604 SSR locus. A combined approach of conventional fiber testing and molecular genotyping was employed to identify marker–trait linkages potentially applicable in cotton breeding programs. An in silico analysis of the BNL1604 genomic region was also performed to gain insights into potential candidate genes associated with the marker. The overall objective was to establish a molecularly guided breeding strategy of next-generation naturally colored cotton with competitive fiber quality.

## 2. Results

### 2.1. Phenotypic Variation in Fiber Quality Traits: Descriptive Statistics and Parental Contrast

The evaluation of 23 cotton genotypes revealed substantial phenotypic variation across all assessed fiber quality traits (Figure 1 and Figure 2). Upper half mean length (UHML) ranged from 21.9 mm, recorded in a brown-fiber line, to 35.4 mm in a white-fiber cultivar, with a population mean of approximately 26.0 mm. Fiber bundle strength varied from 23.3 g·tex^−1^ to 33.0 g·tex^−1^, averaging 26.5 g·tex^−1^. Fiber elongation values ranged between 4.5% and 6.4%, with a mean of 5.4%. Micronaire (an indicator of fiber fineness and maturity) showed the widest variation, from 2.1 (very fine and/or immature fibers) to 6.2 (coarse or thick fibers), with an overall mean of approximately 4.5.

This wide range of fiber properties reflects the intentional genetic diversity of the germplasm set, which included fine-fiber green cotton, coarse-fiber brown cotton, and elite white-fiber cultivars. The variation observed among F_3_ hybrid populations is consistent with expected genetic segregation for fiber quality traits, highlighting the potential for transgressive recombinants and genetic improvement through hybridization.

The elite white-fiber cultivars exhibited significantly superior fiber properties compared with the naturally colored cotton lines. UHML values for the three white cultivars ranged from 33.4 to 35.4 mm, classifying all as “long staple” types. In contrast, the five colored parental lines produced shorter fibers, ranging from 25.9 mm in line L-4017 (brown fiber) to 29.8 mm in line L-4068 (green fiber), with even the best colored line falling 5–6 mm short of the white-fiber standards.

Fiber strength followed a similar trend. The white cultivars C-6580, C-6570, and C-6577 recorded tenacity values of 31.3, 32.1, and 32.7 g·tex^−1^, respectively, were all classified as “strong” according to HVI standards (≥31 g·tex^−1^). In contrast, colored lines ranged from 25.9 g·tex^−1^ (line L-4017) to 29.8 g·tex^−1^ (line L-4068), with only two lines (4068 and L-4099) reaching the “above average” strength category. Line L-4017 was notably the weakest, falling into the “below average” classification.

Differences in fiber elongation between the two groups were less pronounced. White cultivars ranged from 5.1% to 5.6%, whereas colored lines ranged from 4.5% to 5.4%, with most genotypes falling into the “medium” elongation category (4.5–6.0%).

Micronaire values further emphasized differences between the fiber types. All three white cultivars displayed medium micronaire values (4.1–4.4), indicating mature, fine-quality fibers. Among the colored lines, the green-fiber line L-4068 exhibited an exceptionally low micronaire of 2.7, reflecting very fine and likely immature fibers, whereas the brown-fiber line L-4017 had a high value of 5.3, indicating coarse and thick fibers. The remaining colored lines, including L-4083, L-4092, and L-4099, showed intermediate values around 4.6–4.7.

These observations highlight the inherent fiber quality limitations of naturally colored cotton. Even the most promising colored genotype (L-4068) failed to match the white-fiber cultivars in fiber length or strength, despite exhibiting superior fineness. Conversely, the least favorable colored line (4017) combined low fiber strength with coarse fibers. Nonetheless, the observed intra-group variation within the colored cotton germplasm indicates that targeted selection remains a feasible approach for fiber quality improvement.

### 2.2. Performance of F_3_ Hybrids: UHML and Strength

The fifteen F_3_ hybrid populations generally exhibited fiber quality values intermediate between those of their colored and white parents, although considerable variability was observed due to segregation (Figure 3). Several hybrids demonstrated substantial improvements over their colored parents, and some approached the performance levels of the elite white cultivars. These enhancements may partly reflect residual heterosis typical of segregating generations, and further evaluation of stabilized lines will be necessary to confirm their true genetic potential.

Fiber length (UHML) among the F_3_ hybrids ranged from 21.8 mm to 30.5 mm. The longest fibers were observed in the crosses F_3_ (C-6570 × L-4068) (27.5 mm) and F_3_ (C-6577 × L-4099) (27.2 mm), both involving the white parent and C-6570 and C-6577. These UHML values surpassed those of all colored parental lines and considerably reduced the gap to the white standards, although they remained approximately 2–4 mm shorter than the elite white parents, indicating residual heterosis from the F_1_ generation.

In contrast, the shortest UHML values among the hybrids were recorded in crosses involving the white parent C-6577, which itself had slightly shorter fibers (~33.0 mm) compared to C-6577 and C-6580. Specifically, F_3_ (C-6570 × L-4092) (24.8 mm) and F_3_ (C-6577 × L-4017) (24.8 mm) represented the lowest UHML values among the hybrid set, underscoring the influence of parental fiber length on hybrid performance.

Most of the remaining F_3_ hybrid combinations exhibited UHML values in the upper 20 mm range, specifically between 25.0 and 27.0 mm. These results highlight the favorable general combining ability of the white-fiber parents, particularly for fiber length. Notably, crosses involving C-6577—the white cultivar with the longest fiber, consistently produced hybrids with higher UHML compared to those involving C-6570. On average, C-6577 contributed an increase of approximately 1–2 mm in hybrid fiber length relative to C-6570, indicating a favorable parental contribution and strong general combining ability. Collectively, all F_3_ hybrids fell within the “medium staple” category (UHML in the upper 20 mm range), representing a marked improvement over the “short staple” classification assigned to the lowest-performing colored parent (Table 1). These improvements may partly reflect heterosis effects typically observed in segregating generations, contributing to enhanced fiber length and quality performance.

Fiber strength in the F_3_ hybrid populations followed trends similar to those observed for fiber length. The highest strength values were recorded in hybrids derived from crosses involving the white parent C-6577. Specifically, F_3_ (C-6577 × L-4099) achieved a maximum of 28.1 g·tex^−1^, followed closely by F_3_ (C-6577 × L-4092) at 28.0 g·tex^−1^. These results suggest favorable allele complementation between the elite white cultivar and the colored lines (L-4099: light brown; L-4083: brown), enabling the progeny to match the best-performing-colored parents in fiber strength and approach within ~4 g·tex^−1^ of the white parent’s performance. Both hybrids fall within the upper range of the “above average” strength category according to HVI standards (Table 2).

Fiber strength in the F_3_ hybrid populations followed trends similar to those observed for fiber length. The highest strength values were recorded in hybrids derived from crosses involving the white parent C-6577. Specifically, F_3_ (C-6577 × L-4099) achieved a maximum of 29.53 g·tex^−1^, followed closely by F_3_ (C-6577 × L-4083) at 29.34 g·tex^−1^. These results suggest favorable allele complementation between the elite white cultivar and the colored lines (L-4099: light brown; L-4083: brown), enabling the progeny to match the best-performing-colored parents in fiber strength and approach within ~3 g·tex^−1^ of the white parent’s performance. Both hybrids fall within the upper range of the “above average” strength category according to HVI standards.

Most F_3_ hybrids exhibited strength values in the mid- to upper-20s g·tex^−1^, reflecting moderate improvement over their colored parents. Interestingly, the hybrid F_3_ (C-6577 × L-4068), despite being derived from two strong parents, recorded a relatively low strength of 25.96 g·tex^−1^. This may represent a case of negative transgressive segregation, potentially caused by recombination-induced disruption of favorable allelic interactions or epistatic effects. Another hybrid with suboptimal strength was F_3_ (C-6570 × L-4092) (26.33 g·tex^−1^), likely reflecting the influence of the weaker white parent C-6570.

On average, fiber strength in the F_3_ hybrids tended to align more closely with the values of the colored parents than with those of the elite white cultivars, suggesting only partial dominance in the inheritance of this trait. Even the strongest hybrids, reaching approximately 29.5 g·tex^−1^, did not attain the fiber strength levels of the white parents (31–33 g·tex^−1^), indicating that additional selection cycles or advancement to later generations may be necessary to fully recover the superior tenacity characteristic of elite lines. Nevertheless, nearly all hybrid combinations exhibited substantial improvement over the weaker colored parents. Specifically, 13 out of 4 F_3_ hybrids exceeded 25 g·tex^−1^ in fiber strength, clearly outperforming the weakest colored parent (26.7 g·tex^−1^).

### 2.3. Elongation and Micronaire in F_3_ Populations and Breeding Implications

Fiber elongation in the F_3_ populations demonstrated considerable transgressive segregation, with several hybrids surpassing both parental values. The highest elongation rates were recorded in F_3_ (C-6580 × L-4092) at 5.75% and F_3_ (C-6570 × L-4017) at 5.71%. These values represent the upper range typically observed in upland cotton, where elongation above 6% is considered desirable. Notably, both superior hybrids shared the same colored parent (L-4092, light brown), suggesting a favorable genetic interaction in these crosses that enhanced fiber elasticity (Table 3).

Most remaining hybrids exhibited elongation values ranging from 4.84% to 5.5%, overlapping with or marginally exceeding the parental ranges. Importantly, no hybrid showed elongation below 4.5%, indicating that introgression of elite white cotton germplasm did not compromise this trait. In some cases, elongation was enhanced, likely due to complementary gene action between the parents.

Micronaire values among the F_3_ hybrid populations exhibited a broad range, encompassing nearly the entire spectrum observed in the parental genotypes. The lowest micronaire was recorded in F_3_ (C-6580 × L-4068) at 2.6, closely matching its green-fiber parent (L-4068) with a value of 2.9. This indicates that the ultra-fine fiber characteristic of green cotton can be successfully transmitted to and retained in hybrid progeny, even in the presence of a white-fiber genomic background. Another hybrid, F_3_ (C-6570 × L-4068), also displayed a low micronaire (2.9), placing it within the “fine” category (Table 4).

In contrast, several hybrids exhibited elevated micronaire values (>5.0), indicative of coarser fibers. Notable examples include F_3_ (C-6570 × L-4017), F_3_ (C-6570 × L-4092), and F_3_ (C-6570 × L-4099), with values ranging from 5.1 to 5.4. These results likely reflect the inheritance of coarser fiber traits from the respective colored-fiber parents, suggesting non-uniform transmission of fiber fineness alleles across hybrid combinations.

Importantly, the majority of F_3_ hybrids clustered within the medium micronaire category (approximately 4.0–4.9), which is considered optimal for fiber maturity, processing performance, and spinning efficiency. Several hybrids, such as F_3_ (C-6577 × L-4083), F_3_ (C-6577 × L-4099), F_3_ (C-6570 × L-4083), F_3_ (C-6580 × L-4099), and F_3_ (C-6577 × 4017) demonstrated micronaire values (~4.2–4.8) comparable to those of elite white-fiber cultivars. These genotypes are particularly promising as they combine acceptable fiber fineness with concurrent improvements in fiber length and strength.

In summary, the evaluation of the F_3_ populations shows that hybridization between naturally colored and elite white-fiber cotton cultivars enhanced fiber quality traits in the colored cotton background. All hybrid combinations outperformed the weakest colored parent in both fiber length and strength. Several hybrids equaled or exceeded the best colored parent for specific traits. None of the F_3_ hybrids fully matched the elite white cultivars in fiber length and strength simultaneously. However, the performance gap was substantially reduced. On average, the difference declined to within 10–15% of white parent values, compared with the 20–25% deficit observed in the original colored lines.

Significantly, hybridization did not compromise fiber fineness or elongation. Many hybrids retained the desirable fineness of the colored parent. Several also showed increased fiber elongation, indicating transgressive segregation.

Notably, multiple F_3_ lines combined UHML values of 29–30 mm, fiber strength of 28–29 g·tex^−1^, and medium micronaire (~4.5). These results highlight strong potential for developing improved, high-quality, naturally colored cotton cultivars.

### 2.4. Correlations Among Fiber Traits

Pearson correlation analysis across the genotypes revealed several significant relationships (Figure 4). Fiber length (UHML) was found to be strongly positively correlated with fiber strength (r = 0.8, *p* < 0.001). In other words, genotypes with longer fibers tended to also have stronger fibers. This is an encouraging result, as it suggests that the genes improving length and strength may be linked or synergistic. From a breeding perspective, it means selection for longer fiber will generally also improve strength, a coupling of traits that has been noted in upland cotton and is likely due to parallel developmental processes influencing both lengthening and thickening of fibers. Figure 2 illustrates this correlation (longer-fiber hybrids like C-6577 × 4068 also had high strength, whereas short-fiber entries had low strength). Fiber length vs. elongation: a weak positive correlation was observed (r = 0.29, *p* < 0.001). There was a slight tendency for longer fibers to be more elastic, though the relationship was not very tight. Fiber length vs. micronaire: interestingly, a slight negative correlation was found (r = −0.22, *p* < 0.001), indicating that longer fibers were somewhat finer. The effect size is small, but it suggests that in this material, the longest fiber genotypes (which were the white × green hybrids) also had low micronaire, whereas the high-micronaire genotypes (coarse brown cotton) had shorter fiber. This inverse relation may simply reflect the contrasting parent contributions (green cotton provided both long and fine fiber, brown provided short and coarse). It implies that improving fiber length need not come at the expense of fiber fineness—in fact, here the better length came with better fineness, which is favorable for breeding. Fiber strength vs. elongation: there was a moderately strong positive correlation between strength and elongation (r = 0.64, *p* < 0.001). Stronger fibers were often more elastic. This relationship makes intuitive sense because a certain degree of elasticity or extensibility can contribute to toughness (a fiber that elongates slightly will dissipate energy and resist breaking).

The data suggest that no antagonism exists between strength and elongation in these genotypes; on the contrary, improving one may improve the other. Indeed, the white-fiber parental lines exhibited high strength and adequate elongation, while certain hybrids (such as C-6577 × 4092) combined both of these desirable traits at a high level. A very weak negative correlation was detected between fiber strength and micronaire (r = −0.018, *p* = 0.79, n.s.), indicating that the two traits are inherited independently of each other. This is an important point: the fine-fiber green line had lower strength than white cotton due to other factors, not due to its fineness per se; likewise, the coarse brown line had only moderate strength. For breeders, this implies that one can alter fiber fineness (say, make fibers finer to meet market preference) without necessarily hurting strength, as long as the genetic sources of strength are maintained. Elongation vs. micronaire: a positive correlation was observed (r = 0.6, *p* < 0.001). When elasticity is high, the fiber tends to become coarser. This pattern was evident for some brown-fiber progeny: those with high elongation (like some 4092 hybrids) also had above-average micronaire. In summary, the correlation analysis suggests that fiber length, strength, and elongation are positively interrelated in this germplasm—a fortunate scenario where improvement in one should benefit the others. Fiber fineness (micronaire) is largely independent of strength and slightly inversely related to length, indicating that it’s feasible to improve fineness while maintaining strength and perhaps sacrificing a slight bit of length (or vice versa). These relationships are consistent with general observations in upland cotton breeding and indicate no major antagonisms among the key fiber traits, which is advantageous for developing balanced fiber quality.

### 2.5. BNL1604 SSR Marker Polymorphism

PCR screening was performed on 23 research samples utilizing the BNL1604 Simple Sequence Repeat (SSR) marker. As anticipated, PCR amplification demonstrated distinct polymorphism between the parental genotypes. Three alleles of varying sizes, approximately 107 base pairs (bp), 123 bp, and 135 bp, were identified at this locus, as confirmed by the band positions on the gel.

Notably, while white-fibered cultivars exhibited all three alleles, only the 123 bp and 135 bp fragments were amplified in naturally colored fiber lines (e.g., L-4083, L-4068, etc.) (Figure 5). Each genotype produced two to three PCR fragments, indicating that the BNL1604 marker amplifies multiple loci (or heterozygous loci) within the tetraploid cotton genome.

Based on the presence or absence of the three alleles (107 bp, 123 bp, and 135 bp) for the BNL1604 marker across the 23 samples, four allelic combinations (multi-allelic genotypes) were identified. These combinations and their frequencies among the samples were as follows:−107 + 123 + 135 bp (all three alleles present)—observed in 7 genotypes (samples 1, 2, 3, 9, 12, 13, 19);−123 + 135 bp—in 7 genotypes (samples 4, 10, 11, 17, 18, 20, 22);−107 + 123 bp—in 8 genotypes (samples 5, 6, 7, 8, 15, 16, 21, 23);−107 + 135 bp—in 1 genotype (sample 14).

The 123 bp allele was detected in nearly all genotypes (22 out of 23, or approximately 95.6%), suggesting it is a common allele, potentially fixed at one subgenome locus among these cotton genotypes. The ‘123 + 135 bp’ and ‘107 + 123 bp’ allelic combinations were observed at nearly similar medium frequencies (in 7 and 8 genotypes, respectively). Within the ‘107 + 123 bp’ allelic combination (containing the 107 bp allele), positive indicators for both fiber traits (strength and length) were recorded in the F_3_ (C-6577 × L-4099) hybrid combination. It can be concluded that the presence of the 107 bp and 123 bp allelic combination for the BNL1604 marker may be genetically associated with high-quality fiber, specifically fiber strength and length traits, in naturally colored cotton. This suggests its potential future use as a reliable molecular marker for fiber quality selection.

### 2.6. In Silico Chromosomal Localization of BNL1604 in Upland Cotton Genome

The in silico PCR analysis confirmed the chromosomal localization of BNL1604 within both the At and Dt subgenomes of *G. hirsutum*. Specifically, amplified fragments mapped to chromosomes A07 and D07, consistent with previously reported genetic maps. Comparative alignment further supported the presence of homeologous loci across the two subgenomes, reinforcing the reliability of BNL1604 as a marker linked to fiber quality traits (Appendix A, Figure 6).

The genomic regions surrounding the primer-binding sites (~±250 kb) were retrieved and annotated. Several predicted genes with functional relevance to fiber development were identified, including serine/threonine kinases, F-box proteins, and zinc finger transcription factors, which are known to regulate cell elongation, cell wall biosynthesis, and developmental processes.

As illustrated in Figure 5, BNL1604 was localized to both A07 and D07 chromosomes, with physical positions consistent with previously published linkage and QTL mapping studies [32,33,34,35]. Importantly, the A07 localization of BNL1604 overlapped with QTL intervals for fiber length and strength, confirming its reliability as a functional marker for fiber quality improvement.

Comparative analysis of genetic and physical maps demonstrated reasonable concordance between the in silico-derived positions and earlier reports, supporting the potential of BNL1604 as a reliable marker for fiber quality improvement in cotton breeding. These findings underscore the value of integrating in silico mapping with molecular breeding strategies to accelerate genetic enhancement of fiber traits in upland cotton.

Comparative analysis demonstrated reasonable concordance between the in silico-derived positions and previously published QTL and LD-based association mapping studies. This consistency reinforces the reliability of the identified marker–trait associations and highlights their potential application in marker-assisted breeding programs aimed at improving fiber quality and yield in cotton.

### 2.7. Marker–Trait Association Analysis

To assess the effect of the 107 bp allele of the BNL1604 marker on fiber quality traits in F_3_ recombinant genotypes, we conducted a t-test analysis (Table 5). The genotypes were divided into two groups: those with the 107 bp allele present (N = 11) and those without it (N = 4). The analysis results (Table 5) showed a statistically significant correlation (*p* < 0.01) between the presence of the 107 bp allele and all main fiber quality indicators. In the group with this allele, UHML (28.52 mm), strength (29.17 g·tex^−1^), and uniformity (83.88%) were higher, while micronaire was lower (4.18).

The results demonstrate that intraspecific hybridization between naturally colored and elite white-fiber cotton cultivars effectively improved fiber quality traits in segregating populations. F_3_ progenies exhibited significant phenotypic variation and transgressive segregation, with several hybrids approaching elite standards for fiber length, strength, and elongation while retaining natural pigmentation. Micronaire values mostly fell within the acceptable range, indicating that acceptable levels of fiber fineness and maturity can be maintained alongside improvements in other fiber quality traits. Molecular analysis revealed that the SSR marker BNL1604, particularly its 107 bp allele, was consistently associated with superior fiber properties.

## 3. Discussion

This study integrated high-volume instrument (HVI)-based phenotypic assessment with SSR marker genotyping to investigate fiber quality improvement in naturally colored cotton (NCC) and their F_3_ hybrid progenies. The central aim was to combine natural pigmentation with superior fiber quality and evaluate the utility of the marker BNL1604 for marker-assisted selection (MAS) in *Gossypium hirsutum*.

Consistent with earlier studies [4,11], brown- and green-lint NCC lines displayed shorter fibers, lower strength, and in some cases elevated micronaire, likely reflecting linkage drag or pleiotropic effects of pigmentation genes [5,6,7,8]. However, intergroup hybridization with elite white-fiber cultivars (C-6570, C-6577, C-6580) substantially enhanced fiber traits in the F_3_ populations. Several hybrids exhibited transgressive segregation, reducing the performance gap with white cultivars to ~10–15%, compared with the 20–25% deficit observed in the parental NCC lines. Although the number of evaluated genotypes was limited to 15 F_3_ hybrids and three elite cultivars, this experimental design provided sufficient genetic variation and segregation for robust assessment. Thus, the number of lines was not a major limitation in this study.

A key finding of our study is the significant association between the presence of the 107 bp allele and superior fiber quality. Genotypes carrying the 107 bp allele (N = 11) showed significantly higher UHML (28.52 mm) and Strength (29.17 g·tex^−1^) compared to genotypes lacking this allele (N = 4), which had mean UHML of 27.29 mm and Strength of 27.88 g·tex^−1^ (*p* < 0.001 for both traits). This finding confirms earlier QTL and association mapping results [1,19,29] and highlights its potential. The 135 bp allele, frequently detected in advanced breeding material, likely originated from exotic germplasm. While the 123 bp allele itself was nearly fixed (found in 22 of 23 genotypes), the presence of the 107 bp allele was the critical factor associated with high performance. The ability of BNL1604 to discriminate performance groups based on the presence of this 107 bp allele highlights its potential as a predictive marker for early-generation selection.

Importantly, hybridization did not compromise fiber fineness or elongation. Notably, the 107 bp allele group also showed finer (lower) micronaire (4.18 vs. 4.50, *p* < 0.01) and better elongation (7.32% vs. 7.00%, *p* < 0.01). This finding challenges the long-held view that pigmentation genes inherently restrict fiber quality and demonstrates that coloration and performance traits can be genetically compatible.

It is important to note that as *G. hirsutum* is an allotetraploid, the BNL1604 primers likely amplify the target sequence from both homeologous chromosomes (A07 and D07). Consequently, the associations observed in this study represent the combined effect of these loci. Disentangling the specific contributions of each homeolog would require advanced approaches, such as the design of genome-specific primers or amplicon sequencing, which was beyond the scope of this work.

In silico mapping of BNL1604 on chromosome A07 identified nearby genes encoding serine/threonine kinases, F-box proteins, and C2H2-type zinc finger transcription factors—key regulators of cell cycle progression, elongation, and secondary cell wall biosynthesis. These candidate genes provide a plausible mechanistic basis for the observed marker–trait associations and confirm the biological relevance of BNL1604 in fiber development.

From a breeding perspective, integrating phenotypic evaluation with MAS targeting the 107 bp allele of BNL1604 offers a practical approach to accelerate improvement of NCC. Early marker-based screening could identify promising progenies at the seedling stage, minimizing reliance on time-consuming and costly field trials.

The substantial improvements observed in some F_3_ progenies relative to their colored parents may partly reflect residual heterosis and segregation effects inherent to this generation. Therefore, while promising, these results should be considered preliminary, and further evaluation of advanced generations (F_5_ and beyond) will be required to confirm the stability of improved fiber quality traits.

Overall, this work advances the understanding of fiber quality improvement in NCC and provides a robust framework for combining eco-friendly pigmentation with high-end textile performance, addressing both environmental and industrial demands.

## 4. Materials and Methods

### 4.1. Plant Materials

The plant materials used in this study consisted of naturally colored cotton (NCC) lines, elite white-fiber cotton (WFC) cultivars of *Gossypium hirsutum* L., and their F_3_ hybrid progenies derived from intergroup hybridization. Five NCC genotypes with distinct lint pigmentation were selected as pollen parents: L-4083 (brown), L-4017 (brown), L-4099 (light brown), L-4092 (light brown), and Cat. No. L-4068 (green). These colored lines were developed in earlier breeding programs and represent a range of brown and green lint phenotypes.

In total, the study encompassed 23 cotton genotypes, including eight parental genotypes and fifteen F_3_ hybrids (15 genotypes) (Appendix A). The elite white-fiber cultivars C-6580, C-6577, and C-6570 were developed in the Uzbek national cotton breeding program and are widely recognized as standards for superior fiber quality. These cultivars possess extra-long staple fibers (UHML 30–32 mm), high bundle strength (28–30 g·tex^−1^), and optimal micronaire (4.2–4.6), making them suitable as reference parents in hybridization. Their stable performance across environments and previously reported association with fiber quality QTLs justified their use as elite controls in this study.

All plant materials were grown under field conditions in the Tashkent region, Uzbekistan (Kibray; Latitude 41°22′05.3″ N, Longitude 69°24′17.9″ E) during the 2024 growing season. Standard agronomic practices for irrigated cotton were applied throughout the season. Each genotype-whether parental line or F_3_ hybrid- was evaluated under a randomized complete block design (RCBD) with three replications, each plot consisting of 10–15 plants. This design ensured uniform agronomic practices and minimized environmental variation across plots. At physiological maturity, open-pollinated bolls were harvested, and fiber samples were collected for subsequent quality assessment.

The experiment was conducted in three randomized replications using a randomized complete block design. Each plot consisted of 5 rows, 5 m in length, with 0.6 m spacing between rows and 0.1 m spacing between plants. Uniform irrigation, fertilization (N-P-K at recommended rates), application of micronutrients, and pest management were provided. All agronomic practices, including irrigation regime and fertilizer application, were carried out according to the standard technology recommended by the Uzbek Research Institute of Cotton Breeding and Seed Production. Soil fertility was assessed prior to planting to ensure homogeneity, and irrigation was applied uniformly across all plots. Randomization within blocks was employed to reduce bias, thereby ensuring reliable comparisons of fiber quality traits among genotypes.

### 4.2. Hybridization Design and Development of Segregating Populations

Three elite white-fiber cotton cultivars—C-6580, C-6577, and C-6570, developed in Uzbekistan and characterized by superior fiber properties (including high fiber length, strength, and fineness) were selected as female parents. Each white-fiber cultivar was crossed with all five naturally colored cotton (NCC) lines in a complete 3 × 5 factorial mating design, with the white-fiber cultivars used systematically as the female parents. Emasculation and hand-pollination were performed following standard protocols to ensure maternal inheritance from WFCs and to reduce the risk of accidental outcrossing.

This crossing scheme yielded all 15 planned F_1_ hybrid combinations without evidence of hybrid sterility or gross developmental abnormalities. The F_1_ plants were self-pollinated to produce F_2_ progeny, which in turn were advanced by selfing to generate F_3_ populations (Figure 7).

As expected, lint color segregated in segregating generations, while fiber quality traits displayed continuous variation. F_3_ families were prioritized for evaluation because they produce sufficient seed cotton for reliable HVI testing and permit assessment of genetic segregation for fiber traits within families. Representative plants from each F_3_ family were sampled according to a standart randomized block design with three replications, and phenotypic measurements were paired with marker data to support downstream analyses.

### 4.3. Assessment of Fiber Quality Parameters

Fiber quality assessment was performed using a High-Volume Instrument (HVI) system (Uster Technologies AG, Uster, Switzerland). Mature bolls were hand-harvested from each genotype at physiological maturity, and lint was extracted using a laboratory roller gin. For each parental genotype and F_3_ hybrid family, lint from multiple plants was bulked to prepare a representative composite sample (~20 g per genotype). Prior to testing, all samples were conditioned under standardized atmospheric conditions (temperature and relative humidity) for 48 h.

Four key fiber quality parameters were measured on an Uster^®^ HVI 1401 system (Uster Technologies AG, Uster, Switzerland):Upper Half Mean Length (UHML, mm): The average length of the longest 50% of fibers in a sample, serving as a primary indicator of fiber length and spinning performance.Fiber Strength (g·tex^−1^): The breaking strength of a bundle of fibers measured in grams per tex (tex = mass in grams of 1000 m of fiber), reflecting fiber tenacity and its contribution to yarn strength.Fiber Elongation (%): The percentage elongation of a fiber bundle at the break point, representing fiber elasticity and flexibility during mechanical processing.Micronaire (MIC): A dimensionless index integrating fiber fineness and maturity, determined by air permeability of a compressed fiber plug. Lower values indicate finer and/or less mature fibers, while higher values indicate coarser or thicker fibers.

Each sample was tested in triplicate to ensure measurement accuracy and reproducibility. The HVI device was calibrated using USDA-certified cotton standards. In addition to raw data, fiber performance was interpreted based on HVI classification thresholds (e.g., fiber strength ≥ 31 g·tex^−1^ classified as “strong”).

### 4.4. DNA Extraction and Genotyping

Genomic DNA was extracted from young leaf tissue of each cotton genotype. Approximately 100 mg of fresh leaf tissue was collected from greenhouse-grown seedlings to minimize the risk of field dust contamination. DNA isolation followed a modified cetyltrimethylammonium bromide (CTAB) protocol based on Doyle and Doyle, with minor adjustments to optimize yield and purity [36].

#### 4.4.1. PCR Amplification:

The SSR marker BNL1604 was amplified using its published primer sequences: forward 5′-AGAGGGAGTAAAGATTTGGGG-3′ and reverse 5′-TCCAGTTCTTTTTGCCTTGG-3′. Each reaction was performed in a 50 µL volume containing: 1 × PCR buffer (with MgCl_2_), 0.1 mg/mL BSA, 0.2 mM of each dNTP, 0.5 µM of each primer, 2.5 U Taq DNA polymerase, and ~50 ng of template DNA. A touchdown PCR protocol was employed to enhance specificity, consisting of an initial denaturation at 95 °C for 3 min, followed by 45 cycles of 94 °C for 1 min, 50 °C for 1 min, and 72 °C for 2 min, with a final extension at 72 °C for 5 min. Annealing temperatures during the initial cycles were adjusted as necessary. Negative controls (no template DNA) were included in all runs to monitor potential contamination. The SSR markers used in this study, along with their primer sequences and chromosomal positions, are listed in Appendix A.

#### 4.4.2. Gel Electrophoresis and Allele Scoring:

PCR products were separated on 3.5% MetaPhor™ high-resolution agarose gels in 1 × TBE buffer at 5–6 V/cm for approximately 4 h. A 50 bp DNA ladder (Thermo Fisher Scientific, Waltham, MA, USA) served as a molecular size standard. Gels were stained with ethidium bromide (0.5 µg/mL) and visualized under UV illumination. Alleles were identified based on band size (bp) relative to the DNA ladder and confirmed using reference cotton DNA samples with known allele profiles. Alleles were designated by size (e.g., “107 bp”, “123 bp”), and each PCR was repeated at least twice per genotype to verify reproducibility.

Given that *G. hirsutum* is an allotetraploid species (AADD, 2n = 52), multiple bands per SSR locus can occur. The BNL1604 marker typically amplifies loci from both subgenomes, producing one or two bands per genotype depending on homozygosity or heterozygosity. In some cases, additional bands were observed due to sequence divergence between homeologous loci. All distinct bands were recorded, and the presence or absence of the three major alleles was scored in a binary matrix for subsequent data analysis.

### 4.5. In Silico Analysis of the BNL1604 Locus and Comparative Mapping

The published primer sequences of BNL1604 were analyzed using in silico PCR against the *Gossypium hirsutum* reference genome [37] with Unipro UGENE v1.31.0 [38] and NCBI BLAST (version 2.16.0). The marker was anchored to chromosome A07 of the At subgenome, and a homoeologous locus was detected on chromosome D07 of the Dt subgenome [39,40]. A genomic window of approximately ±250 kb flanking the primer binding sites was retrieved for structural annotation. Gene models within this interval were identified using the latest cotton genome annotations and validated through synteny with *Arabidopsis thaliana* to infer candidate genes associated with fiber development.

In addition, virtual PCR products were analyzed to determine the start and end coordinates of amplified chromosomal regions, and results were stored in GB format files to precisely define genomic positions. Chromosomal distribution of the virtual amplicons was visualized using MapChart v2.3 [41], where genomic coordinates of each locus were mapped according to forward primer positions.

For comparative validation, the identified genomic positions were cross-checked with previously published datasets. Linkage disequilibrium (LD)-based association mapping and QTL mapping studies in cotton [1,32,33,34,35,42,43] were used for consistency checks. This comparative analysis confirmed the alignment of BNL1604 with reported QTL regions and facilitated the integration of our results with international genomic databases.

### 4.6. Statistical and Marker–Trait Association Analysis

Descriptive statistics and analysis of variance (ANOVA) were applied to evaluate fiber quality traits [44]. All statistical analyses were conducted using R software (version 4.3.1; R Core Team, Vienna, Austria), employing the packages stats for ANOVA and corrplot for correlation analysis. For each trait, the mean, standard deviation, range, and coefficient of variation (CV) were calculated separately for the NCC parental lines, WFC cultivars, and F_3_ hybrids. One-way ANOVA was performed to assess the effect of genotype on each trait using a balanced dataset of 20 genotypes (5 NCC + 3 WFC + 12 representative F_3_ hybrids).

Pearson’s correlation coefficients (*r*) were computed for all pairwise combinations of the four fiber traits to assess inter-trait relationships. Statistical significance was determined at *p* < 0.05 and *p* < 0.001 using two-tailed *t*-tests. These correlations were used to identify potential trade-offs or co-regulation among fiber properties.

For SSR marker data, the frequency of each BNL1604 allele was calculated as the percentage of genotypes in which the allele was present. As each F_3_ line was derived from a single F_2_ plant and the parental lines were inbred, the observed allele distribution closely reflected true segregation patterns within the germplasm panel.

Associations between BNL1604 alleles and fiber traits were explored using single-marker analysis. Genotypes were grouped based on the presence or absence of specific alleles (e.g., 135 bp or 107 bp), and mean trait values were compared using *t*-tests. While this approach provided preliminary insights into marker–trait relationships, interpretations were made cautiously due to the limited sample size and potential confounding effects of pedigree structure. Robust confirmation of marker–trait associations would require larger, genetically diverse populations and formal QTL mapping approaches.

Genotypes were grouped based on the presence or absence of specific alleles (e.g., 135 bp or 107 bp), and mean trait values were compared using *t*-tests.

## 5. Conclusions

Hybridization between naturally colored cotton lines and elite white-fiber cultivars significantly enhanced key fiber quality traits—UHML, bundle strength, elongation, and micronaire—within the colored cotton background. Several F_3_ hybrids surpassed their pigmented parents and approached the performance of commercial white cultivars, confirming the potential of this approach.

Molecular genotyping with the SSR marker BNL1604 revealed three alleles (107 bp, 123 bp, 135 bp), with the 107 bp allele consistently associated with superior UHML and strength. In silico annotation placed BNL1604 in a genomic region enriched with fiber development-related genes, reinforcing its functional relevance and value for selection.

By integrating BNL1604-based marker-assisted selection with HVI phenotyping, this study identified F_3_ lines that retained natural pigmentation while achieving acceptable fiber quality. These findings provide a practical framework for breeding high-quality, eco-friendly cotton cultivars that combine aesthetic and agronomic value.

## Figures and Tables

**Figure 1 plants-14-03601-f001:**
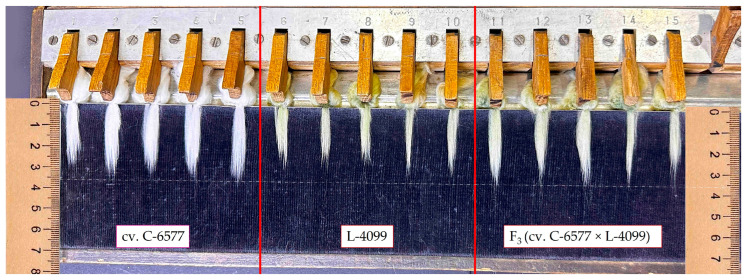
Comparative fiber length assessment in parental genotypes and their hybrid progeny using a comb sorter. The image shows: (**left**) elite white-fiber cultivar C-6577, (**middle**) naturally colored cotton line L-4099 (light brown lint), and (**right**) F_3_ progeny from the cross C-6577 × L-4099. Fibers from each genotype were aligned and measured against a millimeter scale to illustrate differences in staple length and uniformity.

**Figure 2 plants-14-03601-f002:**
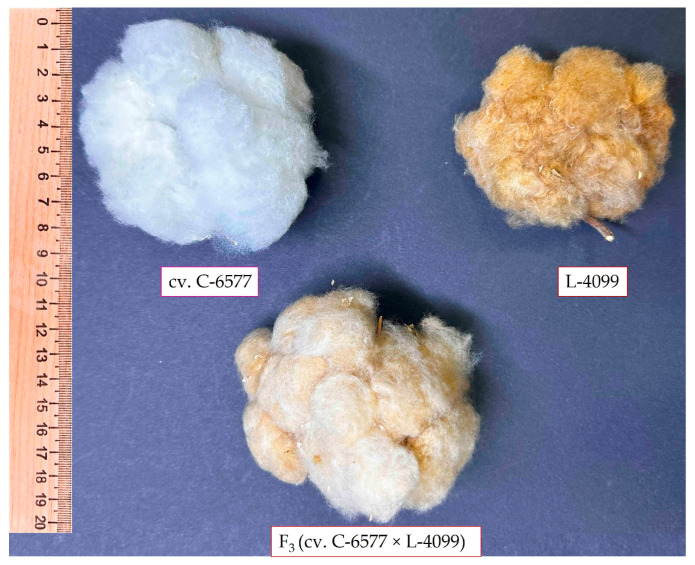
Phenotypic appearance of parental cotton genotypes and their F_3_ hybrid. White fiber of elite cultivar C-6577 (**top left**), brown fiber of naturally colored line L-4099 (**top right**), and light brown fiber of their F_3_ hybrid (C-6577 × L-4099) (**bottom**). The image illustrates fiber color inheritance and variation in the hybrid population.

**Figure 3 plants-14-03601-f003:**
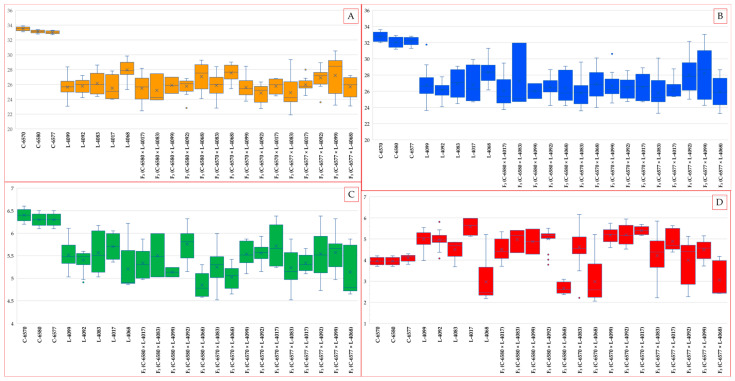
Boxplot distribution of fiber quality parameters in parental genotypes and their F_3_ hybrid progenies: (**A**) Upper Half Mean Length (UHML, mm), (**B**) Fiber Strength (g·tex^−1^), (**C**) Fiber Elongation (%), and (**D**) Micronaire. Data were obtained from High-Volume Instrument (HVI) analysis for elite white-fiber cultivars (C-6570, C-6580, C-6577), naturally colored cotton lines (L-4099, L-4092, L-4083, L-4017, L-4068), and F_3_ progenies from respective crosses. Box edges represent the interquartile range (IQR), horizontal lines within boxes indicate the median, whiskers denote 1.5× IQR, and points represent outliers. The plots illustrate variation among genotypes, transgressive segregation in hybrids, and trait differences between white and colored cotton backgrounds.

**Figure 4 plants-14-03601-f004:**
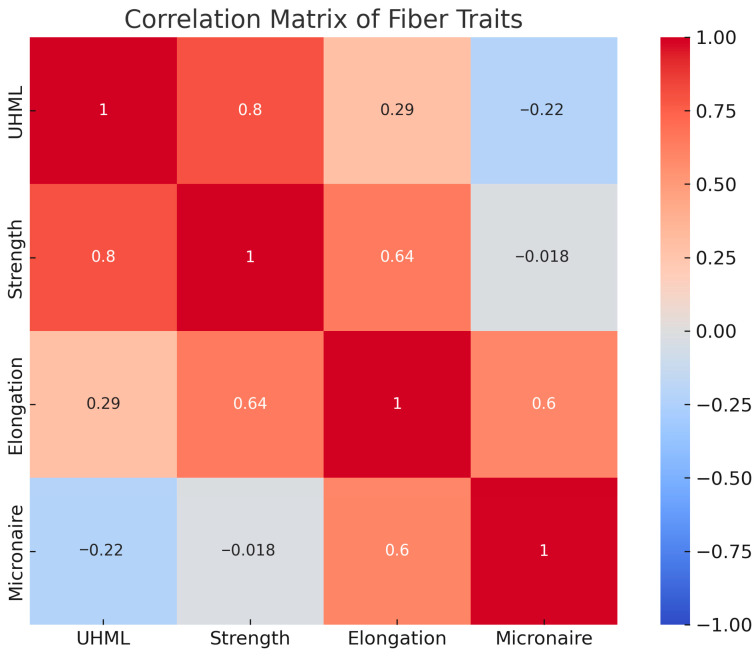
Correlation matrix among fiber quality traits in *Gossypium hirsutum* genotypes, including UHML (upper-half mean length), fiber strength, elongation, and micronaire. Pearson’s correlation coefficients are color-coded from −1 (blue, negative correlation) to +1 (red, positive correlation). Significant positive associations were observed between UHML and strength (r = 0.80), and between strength and elongation (r = 0.64), while micronaire was largely independent.

**Figure 5 plants-14-03601-f005:**
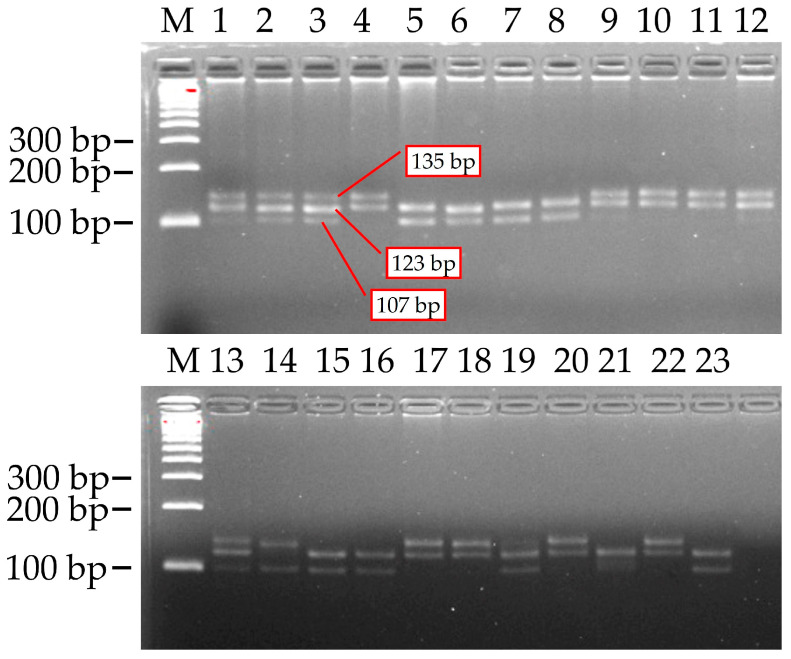
PCR amplification results of cotton genotypes using the BNL1604 marker. M) 100 bp molecular weight marker (DNA ladder); 1–23—Cotton genotypes analyzed: (1) C-6580; (2) C-6570; (3) C-6577; (4) L-4083; (5) F_3_C-6580 × L-4017; (6) F_3_C-6580 × L-4083; (7) F_3_C-6580 × L-4092; (8) F_3_C-6580 × L-4068; (9) L-4068; (10) L-4017; (11) L-4099; (12) L-4092; (13) F_3_C-6570 × L-4099; (14) F_3_C-6580 × L-4099; (15) F_3_C-6570 × L-4092; (16) F_3_C-6577 × L-4017; (17) F_3_C-6570 × L-4083; (18) F_3_C-6570 × L-4068; (19) F_3_C-6577 × L-4092; (20) F_3_C-6577 × L-4083; (21) F_3_C-6577 × L-4099; (22) F_3_C-6577 × L-4068; (23) F_3_C-6570 × L-4017.

**Figure 6 plants-14-03601-f006:**
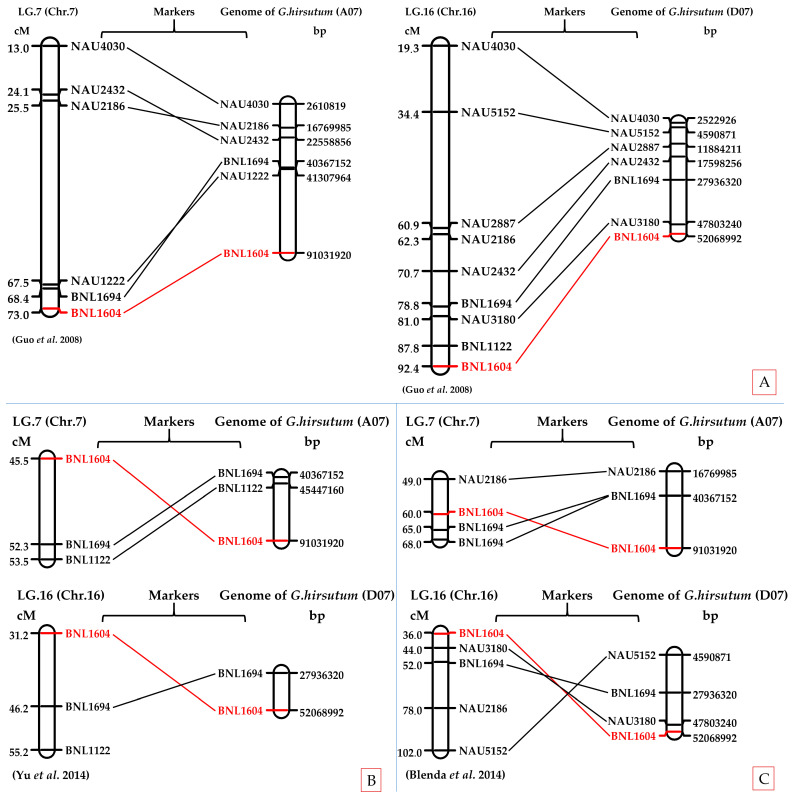
Comparative genetic and physical mapping of the SSR marker BNL1604 in the cotton genome (*Gossypium hirsutum* L.). Linkage group positions (**left**) are based on published genetic maps ((**A**): [33]; (**B**): [34]; (**C**): [35]), while physical positions (**right**) were determined through in silico PCR analysis of the *G. hirsutum* genome. The BNL1604 locus was consistently mapped to chromosomes A07 and D07 (LG7 and LG16), overlapping with QTL regions previously reported for fiber length and strength. Other markers (e.g., NAU4030, NAU2432, NAU2186, NAU5152, NAU2887, NAU3180, NAU1222, NAU2995, BNL1694, and BNL1122) are shown for comparative reference. The repeated confirmation of BNL1604 across multiple maps highlights its reliability as a diagnostic marker for fiber quality improvement in upland cotton.

**Figure 7 plants-14-03601-f007:**
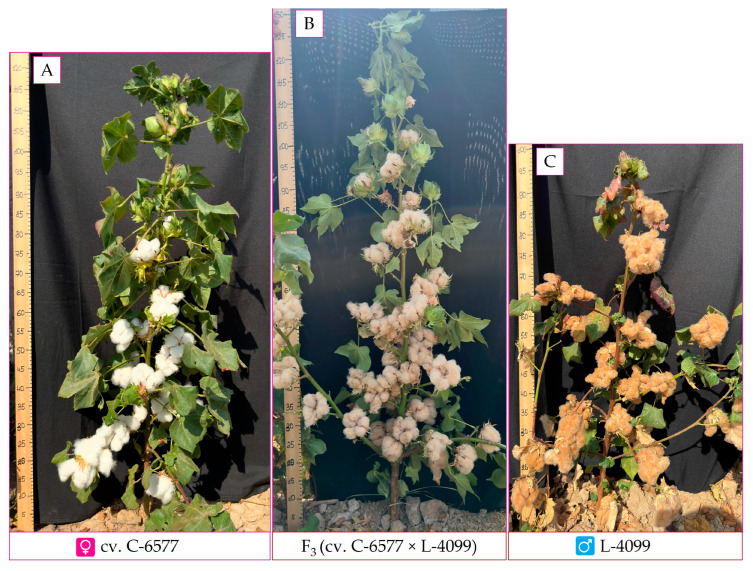
Parental genotypes and their F_3_ hybrid. (**A**)—C-6577; (**B**)—F_3_ hybrid progeny; (**C**)—L-4099.

**Table 1 plants-14-03601-t001:** Fiber length (UHML) in parental lines and F_3_ hybrid combinations.

No.	Samples	Range	$\bar{x}\;\pm$ S $\bar{x}$	*S*	*V*%
Parental genotypes					
1	C-6570	33.1–33.9	33.4 ± 0.13	0.30	0.84
2	C-6580	33.1–33.4	33.2 ± 0.11	0.29	0.60
3	C-6577	32.7–33.3	33.0 ± 0.08	0.21	0.60
4	L-4099	23.0–28.3	25.6 ± 0.27	1.27	4.86
5	L-4092	24.2–27.2	25.7 ± 0.20	0.90	3.52
6	L-4083	24.3–28.6	26.1 ± 0.53	1.52	5.84
7	L-4017	24.1–27.8	25.5 ± 0.88	1.76	6.90
8	L-4068	25.3–29.8	27.9 ± 0.43	1.30	4.67
F_3_ hybrids					
9	F_3_ (C-6580 × L-4017)	22.4–28.1	25.4 ± 0.41	1.71	6.74
10	F_3_ (C-6580 × L-4083)	23.9–27.4	25.2 ± 1.12	1.95	7.73
11	F_3_ (C-6580 × L-4099)	24.7–26.9	25.8 ± 1.10	1.56	6.12
12	F_3_ (C-6580 × L-4092)	22.8–26.7	25.7 ± 0.24	1.00	3.92
13	F_3_ (C-6580 × L-4068)	24.0–29.2	27.0 ± 0.64	1.83	6.80
14	F_3_ (C-6570 × L-4083)	22.8–28.3	25.8 ± 0.29	1.52	5.93
15	F_3_ (C-6570 × L-4068)	25.4–29.0	27.5 ± 0.41	1.17	4.28
16	F_3_ (C-6570 × L-4099)	23.7–28.4	25.5 ± 0.42	1.33	5.24
17	F_3_ (C-6570 × L-4092)	22.7–26.3	24.8 ± 0.43	1.24	5.00
18	F_3_ (C-6570 × L-4017)	24.4–26.7	25.7 ± 0.46	1.03	4.03
19	F_3_ (C-6577 × L-4083)	21.8–29.3	24.8 ± 0.48	1.98	8.00
20	F_3_ (C-6577 × L-4017)	24.5–27.9	25.9 ± 0.32	0.92	3.57
21	F_3_ (C-6577 × L-4092)	23.6–28.9	26.8 ± 0.45	1.42	5.42
22	F_3_ (C-6577 × L-4099)	23.2–30.5	27.2 ± 0.84	2.53	9.32
23	F_3_ (C-6577 × L-4068)	23.0–27.2	25.6 ± 0.70	1.58	6.20

**Table 2 plants-14-03601-t002:** Fiber strength in parental lines and F_3_ hybrid combinations.

No.	Samples	Range	$\bar{x}\;\pm$ S $\bar{x}$	*S*	*V*%
Parental genotypes					
1	C-6570	32.0–33.6	32.8 ± 0.27	0.62	1.59
2	C-6580	31.2–32.9	32.0 ± 0.22	0.59	2.02
3	C-6577	31.3–32.8	32.1 ± 0.22	0.54	1.70
4	L-4099	23.6–31.7	26.7 ± 0.39	1.81	6.79
5	L-4092	24.0–27.7	26.1 ± 0.22	0.99	3.80
6	L-4083	24.4–29.0	26.9 ± 0.62	1.77	6.60
7	L-4017	24.6–29.9	26.7 ± 1.18	2.36	8.86
8	L-4068	26.1–31.3	28.3 ± 0.50	1.52	5.37
F_3_ hybrids					
9	F_3_ (C-6580 × L-4017)	23.7–29.4	26.0 ± 0.43	1.77	6.83
10	F_3_ (C-6580 × L-4083)	24.7–31.9	27.2 ± 2.34	4.06	14.9
11	F_3_ (C-6580 × L-4099)	25.0–26.9	26.0 ± 0.93	1.32	5.01
12	F_3_ (C-6580 × L-4092)	24.2–28.6	26.5 ± 0.25	1.06	4.02
13	F_3_ (C-6580 × L-4068)	24.2–29.0	26.3 ± 0.68	1.94	7.39
14	F_3_ (C-6570 × L-4083)	23.5–29.6	25.7 ± 0.27	1.41	5.53
15	F_3_ (C-6570 × L-4068)	24.0–30.1	26.9 ± 0.67	1.91	7.12
16	F_3_ (C-6570 × L-4099)	24.5–30.6	26.6 ± 0.54	1.73	6.52
17	F_3_ (C-6570 × L-4092)	24.7–28.5	26.3 ± 0.46	1.30	4.95
18	F_3_ (C-6570 × L-4017)	24.7–28.9	26.4 ± 0.77	1.73	6.58
19	F_3_ (C-6577 × L-4083)	23.2–30.0	25.8 ± 0.43	1.80	7.04
20	F_3_ (C-6577 × L-4017)	25.3–28.7	26.2 ± 0.38	1.15	4.39
21	F_3_ (C-6577 × L-4092)	25.0–32.1	28.0 ± 0.72	2.28	8.14
22	F_3_ (C-6577 × L-4099)	24.2–33.0	28.1 ± 1.09	3.27	11.6
23	F_3_ (C-6577 × L-4068)	23.2–28.6	25.9 ± 0.87	1.95	7.55

**Table 3 plants-14-03601-t003:** Fiber elongation in parental lines and F_3_ hybrid combinations.

No.	Samples	Range	$\bar{x}\;\pm$ S $\bar{x}$	*S*	*V*%
Parental genotypes					
1	C-6570	6.2–6.6	6.36 ± 0.05	0.11	2.20
2	C-6580	6.1–6.5	6.34 ± 0.06	0.17	2.24
3	C-6577	6.1–6.5	6.3 ± 0.05	0.14	2.24
4	L-4099	5.03–6.11	5.51 ± 0.06	0.30	5.44
5	L-4092	4.91–5.6	5.40 ± 0.04	0.19	3.55
6	L-4083	5.03–6.17	5.56 ± 0.16	0.45	8.32
7	L-4017	5.36–6.05	5.70 ± 0.14	0.29	5.54
8	L-4068	4.86–6.22	5.20 ± 0.16	0.49	9.61
F_3_ hybrids					
9	F_3_ (C-6580 × L-4017)	4.97–5.87	5.33 ± 0.07	0.28	5.34
10	F_3_ (C-6580 × L-4083)	5.03–5.99	5.5 ± 0.27	0.48	9.09
11	F_3_ (C-6580 × L-4099)	5.03–5.24	5.13 ± 0.10	0.14	9.09
12	F_3_ (C-6580 × L-4092)	5.15–6.32	5.75 ± 0.07	0.31	5.49
13	F_3_ (C-6580 × L-4068)	4.58–5.15	4.84 ± 0.09	0.27	5.52
14	F_3_ (C-6570 × L-4083)	4.52–5.99	5.24 ± 0.06	0.34	6.61
15	F_3_ (C-6570 × L-4068)	4.65–5.42	5.02 ± 0.09	0.26	5.32
16	F_3_ (C-6570 × L-4099)	5.1–5.87	5.53 ± 0.08	0.25	4.66
17	F_3_ (C-6570 × L-4092)	5.15–5.93	5.55 ± 0.10	0.22	4.30
18	F_3_ (C-6570 × L-4017)	5.24–6.38	5.71 ± 0.21	0.47	8.30
19	F_3_ (C-6577 × L-4083)	4.52–5.87	5.23 ± 0.08	0.36	6.92
20	F_3_ (C-6577 × L-4017)	5.1–5.66	5.33 ± 0.06	0.19	3.63
21	F_3_ (C-6577 × L-4092)	4.37–6.38	5.54 ± 0.16	0.51	8.92
22	F_3_ (C-6577 × L-4099)	4.97–6.32	5.57 ± 0.12	0.38	6.95
23	F_3_ (C-6577 × L-4068)	4.65–5.87	5.14 ± 0.24	0.55	10.6

**Table 4 plants-14-03601-t004:** Micronaire in parental lines and F_3_ hybrid combinations.

No.	Samples	Range	$\bar{x}\;\pm$ S $\bar{x}$	*S*	*V*%
Parental genotypes					
1	C-6570	3.7–4.2	3.98 ± 0.07	0.15	5.12
2	C-6580	3.7–4.2	3.98 ± 0.07	0.19	5.02
3	C-6577	3.8–4.3	4.1 ± 0.07	0.17	4.87
4	L-4099	3.98–5.54	4.99 ± 0.08	0.39	8.01
5	L-4092	4.08–5.34	4.96 ± 0.08	0.37	7.45
6	L-4083	3.68–4.83	4.50 ± 0.15	0.42	9.57
7	L-4017	5.13–5.98	5.59 ± 0.22	0.44	8.00
8	L-4068	2.3–5.21	2.96 ± 0.37	1.11	37.7
F_3_ hybrids					
9	F_3_ (C-6580 × L-4017)	3.71–5.34	4.50 ± 0.12	0.53	11.7
10	F_3_ (C-6580 × L-4083)	4.35–5.41	4.97 ± 0.32	0.54	11.0
11	F_3_ (C-6580 × L-4099)	4.28–5.47	4.87 ± 0.59	0.83	17.1
12	F_3_ (C-6580 × L-4092)	3.78–5.51	5.01 ± 0.12	0.50	10.1
13	F_3_ (C-6580 × L-4068)	2.36–3.39	2.66 ± 0.09	0.27	10.0
14	F_3_ (C-6570 × L-4083)	2.21–6.16	4.62 ± 0.16	0.82	17.9
15	F_3_ (C-6570 × L-4068)	2.19–5.21	2.97 ± 0.39	1.21	37.7
16	F_3_ (C-6570 × L-4099)	4.59–5.39	5.18 ± 0.12	0.38	7.33
17	F_3_ (C-6570 × L-4092)	4.52–5.95	5.19 ± 0.17	0.49	9.49
18	F_3_ (C-6570 × L-4017)	5.17–5.69	5.4 ± 0.09	0.22	4.14
19	F_3_ (C-6577 × L-4083)	2.21–5.08	4.21 ± 0.21	0.89	42.4
20	F_3_ (C-6577 × L-4017)	4.38–5.63	4.88 ± 0.16	0.50	10.2
21	F_3_ (C-6577 × L-4092)	2.26–5.13	3.99 ± 0.33	1.05	26.5
22	F_3_ (C-6577 × L-4099)	3.72–5.15	4.49 ± 0.15	0.47	10.5
23	F_3_ (C-6577 × L-4068)	2.42–4.18	3.05 ± 0.38	0.85	27.9

**Table 5 plants-14-03601-t005:** Marker–trait association (*t*-test) analysis comparing F_3_ hybrid genotypes with and without the BNL1604 107 bp allele.

No.	Trait	Genotype Group (Based on BNL1604)	No. of F_3_ Hybrids	Mean ± SD	*p*-Value
1.	UHML (mm)	With 107 bp allele	11	28.52 ± 0.70	<0.001
		Without 107 bp allele	4	27.29 ± 0.12	
2.	Strength (g·tex^−1^)	With 107 bp allele	11	29.17 ± 0.61	<0.001
		Without 107 bp allele	4	27.88 ± 0.17	
3.	Micronaire (Mic)	With 107 bp allele	11	4.18 ± 0.13	<0.01
		Without 107 bp allele	4	4.50 ± 0.08	
4.	Elongation (Elg, %)	With 107 bp allele	11	7.32 ± 0.19	<0.01
		Without 107 bp allele	4	7.00 ± 0.08	

## Data Availability

The original contributions presented in this study are included in the article/Appendix A. Further inquiries can be directed to the corresponding authors.

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
