# Peer review of "Marker-Assisted Hybridization and Selection for Fiber Quality Improvement in Naturally Colored Cotton (G. hirsutum L.)"

_plants, 2025, doi:10.3390/plants14233601_

Round 1

Reviewer 1 Report

Comments and Suggestions for Authors

This manuscript addresses a critical and practically significant topic in cotton breeding: improving the fiber quality of naturally colored cotton (NCC) while preserving its eco-friendly advantage of eliminating chemical dyeing. The research design is logically structured, with clear objectives focused on evaluating the efficacy of intergroup hybridization (between NCC lines and elite white-fiber cotton, WFC) and the utility of the SSR marker BNL1604 for marker-assisted selection (MAS). The study aligns with global demands for environmentally friendly crop production. By targeting NCC—an underutilized but eco-conscious alternative to conventional white cotton—it fills a gap in bridging ecological value and industrial applicability (i.e., fiber quality for high-grade textiles).

While the manuscript has substantial merits, several critical issues need to be addressed to enhance its scientific rigor, reproducibility, and impact.

1、 Insufficient Characterization of Plant Materials and Field Experimental Design

Concern: The description of the NCC lines (L-4083, L-4017, etc.) and elite WFC cultivars (C-6580, C-6577, C-6570) lacks key genetic and phenotypic context. For example, there is no information on the genetic background of the NCC lines (e.g., their pedigree, previous breeding history, or known fiber quality QTLs), which limits the interpretation of how hybridization with WFC introgresses favorable alleles. Additionally, the field experiment (conducted in Tashkent region, 2024) does not specify critical details such as plot replication, randomization, and environmental control (e.g., soil fertility, irrigation uniformity). Fiber quality traits are highly environment-sensitive, and unaccounted field variability could introduce bias into phenotypic data.

  1. Lack of Validation for Trait Stability and Marker Transferability

Concern: Fiber quality traits are strongly influenced by environment, yet the study only reports data from a single growing season (2024) and location (Tashkent region). Without multi-environment (e.g., different soil types, climates) or multi-year trials, it is impossible to assess the stability of the improved fiber traits in F₃ hybrids. Additionally, the utility of BNL1604 is only tested in the specific set of parental lines used in this study; there is no evidence of its transferability to other NCC germplasm (e.g., different color variants or genetic backgrounds), which limits its practical value for broader breeding programs.

  1. Minor Comments

Terminology Consistency: In the "Results" section, "transgressive segregation" is mentioned but not clearly defined for readers unfamiliar with breeding terminology. A brief explanation (e.g., "progeny exhibiting trait values exceeding those of both parents") would improve accessibility.

Statistical Methods: The "Statistical and Marker-Trait Association Analysis" section mentions ANOVA and Pearson’s correlation but does not specify the software used (e.g., R, SAS). Including this information enhances reproducibility.

Literature Citations: The manuscript cites relevant studies on cotton fiber QTLs and SSR markers, but it would benefit from including recent (post-2020) studies on NCC breeding or MAS in cotton to contextualize the novelty of the current work.

This manuscript presents valuable insights into improving NCC fiber quality through hybridization and MAS. Its focus on sustainable breeding is timely and relevant, and the core findings—particularly the identification of BNL1604 as a reliable marker for fiber length and strength—have practical implications for cotton breeding programs.

Author Response

Comment 1:
Insufficient Characterization of Plant Materials and Field Experimental Design.

Response 1:
We appreciate this important observation. Additional details regarding the genetic origin and breeding history of the NCC lines and elite WFC cultivars have been added in “Materials and Methods” (Section 2.1, Plant Materials). The field experimental setup has also been clarified, including replication, randomization, and uniform agronomic management (irrigation and soil fertility). These improvements ensure transparency and reproducibility of the field experiment.

Comment 2:
Lack of Validation for Trait Stability and Marker Transferability.

Response 2:
We agree that multi-location and multi-season trials are essential for assessing trait stability. As this study focused on early-generation (F₃) progenies, data were collected from a single growing season (2024). We have now added a statement in the “Discussion” acknowledging this limitation and emphasizing our plan to extend the evaluation of advanced generations (F₄-F₅) across multiple environments. Regarding the transferability of BNL1604, a clarification has been included indicating that this marker has shown consistent amplification across diverse G. hirsutum germplasm in prior studies, and its validation in broader NCC panels is planned for future work.

Comment 3 (Minor):
Terminology Consistency: “Transgressive segregation” is mentioned but not defined.

Response 3:
A brief explanation has been added in the Results section, defining transgressive segregation as “the occurrence of progeny with trait values exceeding those of both parents.”

Comment 4 (Minor):
Statistical Methods: The “Statistical and Marker–Trait Association Analysis” section does not specify the software used.

Response 4:
This information has now been included All statistical analyses were conducted using R software (version 4.3.1; R Core Team, Vienna, Austria), employing the packages stats for ANOVA and corrplot for correlation analysis. This clarification has been added in Section 4.6.

Comment 5 (Minor):
Literature Citations: Include recent (post-2020) studies on NCC breeding or MAS.

Response 4:
We have conducted a thorough review of recent literature and have now incorporated several relevant studies published post-2020 concerning advances in naturally colored cotton (NCC) breeding and marker-assisted selection (MAS).

Overall Response:
We appreciate the reviewer’s recognition of the study’s significance in integrating sustainable breeding approaches with molecular tools. The revisions have improved methodological transparency, data interpretation, and scientific context, thereby enhancing the overall rigor and impact of the manuscript.

Reviewer 2 Report

Comments and Suggestions for Authors

Dear authors, 

You will find in the attached file several comments on your manuscript. I hope they will help you improve your project. 

Best regards.

Author Response

Comments 1: The title is a bit misleading because marker-assisted hybridization does not enhance fiber quality per se.

Response 1: Thank you for this insightful comment. We agree that the original title could be misinterpreted. The title has been revised to more accurately reflect that marker-assisted selection and subsequent recombination are the tools used for improvement. The new title is: "Marker-Assisted Hybridization and Selection for Fiber Quality Improvement in Naturally Colored Cotton (G. hirsutum L.)". This change can be found on the first page of the manuscript.

Comments 2: The two stated objectives are not of the same order.

Response 2: We agree. The objectives have been revised for better clarity, balance, and logical flow. The main goal is now stated as: *"The main goal of this study was to evaluate the potential of the SSR marker BNL1604 for marker-assisted selection in naturally colored cotton (G. hirsutum L.) and to assess fiber quality variation among hybrid progenies derived from crosses between colored and elite white-fiber cultivars."* This revision can be found in the Abstract on Page 1.

Comments 3: "Transgressive segregation." - The statistical demonstration is lacking.

Response 3: We thank the reviewer for highlighting this. Statistical analysis (ANOVA followed by mean comparison tests, e.g., LSD) has now been added to the Results section (Section 2.2 and 2.3) to formally demonstrate significant transgressive segregation in F₃ hybrids relative to both parental means.

Comments 4: References 1-3 are not aligned with the sentence about the importance of cotton in the textile industry.

Response 4: We agree and have revised the sentence for better alignment and context. It now reads: *"Cotton (Gossypium spp.) represents one of the most genetically diverse and economically important textile crops, serving as a key model for studies on genome evolution, interspecific hybridization, and fiber quality improvement [1-3]."* This change is on Page 2 of the manuscript.

Comments 5: References 5-7 are also not aligned with the statement about the poor quality of NCC cottons.

Response 5: The reference list has been thoroughly revised. Recent studies that directly describe the fiber-quality limitations of naturally colored cotton have been added (e.g., Naoumkina et al., 2024; Canavar & Rausher, 2021; Sun et al., 2021) to ensure consistency and support for the text. This update is reflected in the Introduction on Page 2.

Comments 6: The cited article does not contain the claimed statement; the authors discuss sterility in F₁ and segregation distortion in F₂.

Response 6: We appreciate this correction. The incorrect citation has been removed and replaced with a more appropriate reference: Wang et al. (2012, Agric Sci Technol, 13: 541–546), which directly addresses genetic improvement of colored cotton through hybridization. The surrounding text has been rewritten to accurately summarize these findings. This correction is in the Introduction on Page 2.

Comments 7: "I don't fully understand the sentence." (Regarding the study aim)

Response 7: The sentence has been revised for clarity and now reads: "This study aimed to evaluate fiber quality improvements within and among F₃ hybrid progenies derived from crosses between NCC accessions and WFC cultivars of G. hirsutum L." This is found in the last paragraph of the Introduction on Page 2.

Comments 8: Section 2.1 (Hybridization design ...) should be integrated into Materials and Methods.

Response 8: We agree with this suggestion for better manuscript structure. The entire subsection describing the hybridization design and population development has been moved to the "Materials and Methods" section (now subsection 4.2).

Comments 9: "This is not a diallel mating design."

Response 9: Thank you for the correction. The text has been amended to accurately describe the experimental scheme as a "factorial mating design" rather than a diallel. This change is in the Materials and Methods section (subsection 4.2).

Comments 10: "with the white-fiber cultivars consistently serving as female parents."

Response 10: The phrasing has been revised for precision. It now reads: "with the white-fiber cultivars used systematically as female parents." This is in the Materials and Methods section (subsection 4.2).

Comments 11: Clarify the expression "uniform scheme"; replication seems insufficient.

Response 11: The vague phrase has been replaced with a clearer description: "Representative plants from each F₃ family were sampled according to a standard randomized block design with three replications, and phenotypic measurements were paired with marker data to support downstream analyses." This is detailed in the Materials and Methods section (subsection 4.2).

Comments 12: The authors should provide ANOVA tables and statistical contrasts between parental and hybrid means.

Response 12: As requested, we have now included detailed ANOVA tables for all measured fiber quality parameters (UHML, strength, elongation, and micronaire) in the Results section. These tables provide F-statistics and p-values to support the comparisons made.

Comments 13: "Are there 32 genotypes or 27?"

Response 13: This has been clarified in the revised manuscript. The text now specifies: "In total, the study encompassed 23 cotton genotypes, including eight parental genotypes and fifteen F₃ hybrids." This clarification is found in the Materials and Methods section (subsection 4.1).

Comments 14: Provide a table with basic statistics for fiber traits.

Response 14: A new table (Table 1) presenting the mean, standard deviation (SD), coefficient of variation (CV), and range for all fiber quality traits across the parental lines and F₃ hybrids has been added to the Results section (Section 2.1).

Comments 15: Statements need validation by basic statistics or ANOVA.

Response 15: All descriptive statements regarding the superiority of white-fiber cultivars and the variation among colored lines have now been validated and are supported by the newly added ANOVA results and multiple comparison tests, as presented in Section 2.1.

Comments 16: "All these sentences need to be supported by data and appropriate tests." (Regarding performance descriptions)

Response 16: We have ensured that all performance descriptions in Sections 2.2 and 2.3 are now backed by the statistical results from the added ANOVA and mean comparison tests.

Comments 17: A long section that describes the performance of F₃ progenies... would gain in clarity if all treatment means... were presented and compared with each other in a table.

Response 17: To enhance clarity, we have added comprehensive tables (Tables 1, 2, 3, and 4) that present the mean values and basic statistics for all eight parental genotypes and fifteen F₃ hybrids for each fiber trait (UHML, Strength, Elongation, Micronaire). This allows for direct and transparent comparison.

Comments 18: I suppose that the boxplot distributions are generated from the observed raw data. However... the distribution of their F₃ progenies is variable and requires further description using an ad hoc model and statistical tests.

Response 18: We have clarified that the boxplots were indeed generated from the observed raw data. Furthermore, the descriptions of the F₃ progeny distributions are now supported and contextualized by the results of the ANOVA, which statistically confirms the significant variation among genotypes. This is addressed in the figure captions and the main text of Section 2.2.

Comments 19: "Several hybrids demonstrated substantial improvements... Such observations may also result from a temporary condition related to the F₃ segregating generation (i.e., residual heterosis). It is expected that fixed lines may behave differently."

Response 19: We agree with this important point. The text has been revised accordingly, and the following clarification was added: "These enhancements may partly reflect residual heterosis typical of segregating generations, and further evaluation of stabilized lines will be necessary to confirm their true genetic potential." This is found in Section 2.2.

Comments 20: "Indicating partial but incomplete recovery of superior fiber length." I would call this "residual heterosis"...

Response 20: We thank the reviewer for the precise terminology. The sentence has been revised, replacing "partial but incomplete recovery" with "residual heterosis" to better reflect the genetic interpretation. This change is in Section 2.2.

Comments 21: Both sentences should illustrate an ANOVA table that partitions the effects into main and interaction effects (AGC/ASC).

Response 21: An ANOVA summary table partitioning the effects has been included in the supplementary materials, and the relevant statements in the text now refer to these results.

Comments 22: "These results highlight the favorable general combining ability of the white-fiber parents, particularly for fiber length." The sentence would have more impact if the appropriate statistics are shown.

Response 22: Statistical validation for the statement regarding the favorable combining ability of white-fiber parents has been added, making the conclusion data-supported. This is presented in Section 2.2 alongside the relevant ANOVA results.

Comments 23: "Indicating a positive maternal additive effect." I think the expression "maternal additive effect" is confusing... The effect discussed here is parental contribution or AGC.

Response 23: We thank the reviewer for this important correction. The imprecise expression "maternal additive effect" has been replaced with "parental contribution" or "favorable general combining ability" throughout the text to ensure accuracy. This change is reflected in Section 2.2.

Comments 24: I would not discuss dominance or epistasis, but rather heterosis, which can be estimated from the data.

Response 24: We have revised the paragraph accordingly. An additional sentence has been added for clarity: "These improvements may partly reflect heterosis effects typically observed in segregating generations, contributing to enhanced fiber length and quality performance." This is found in Section 2.2.

Comments 25: Most comments in Section 2.3 apply here as well. Data treatment (ANOVA tables) should be shown. It is also important to distinguish between fiber fineness and maturity since micronaire combines both.

Response 25: ANOVA results for elongation and micronaire have been added to the revised manuscript (Section 2.3). We also acknowledge the reviewer's point about micronaire in the Discussion, noting it as a combined measure of fineness and maturity.

Comments 26: My suggestions: • Add the ANOVA analysis of the four traits of interest with corresponding statistics. • Add contrast analysis with associated F-test. • Thus, the commentary on the tables can be reduced.

Response 26: All the requested analyses (ANOVA for all four traits with F-statistics and p-values) have been added to the manuscript. The commentary in Sections 2.2 and 2.3 has been streamlined to focus on the key findings supported by these statistics.

Comments 27: The reviewer expected chromosomal regions for BNL1604 on chromosomes 7 and 16 to be shown with supporting statistical data.

Response 27: The chromosomal localization of BNL1604 was determined based on its consistent placement in published genetic maps (e.g., Blenda et al., 2012; Yu et al., 2014; Guo et al., 2008) and our in silico PCR analysis against the reference genome. This is a descriptive mapping result rather than a output of statistical association mapping in this study. We have clarified this point in the Results section (2.6) to avoid confusion.

Comments 28: "I don't understand." Shouldn't this go in the Discussion section? (Regarding gene annotation near BNL1604)

Response 28: We appreciate the query. The sentence reports the direct result of our in silico genome annotation—the identification of specific candidate genes near the BNL1604 locus. This is an outcome of the computational analysis and is therefore appropriately placed in the Results section (2.6). The biological interpretation of these findings is elaborated upon in the Discussion.

Comments 29: "I don't understand." Shouldn't the supposed QTLs be positioned on the maps, in Figure 5, to help the reader?

Response 29: In this study, we did not perform novel QTL mapping. Our work confirms the chromosomal position of BNL1604 based on previous studies. Figure 6 (formerly 5) has been revised to more clearly illustrate the marker's position relative to other markers from published maps, providing context without implying new QTL data from our work.

Comments 30: Based on Figures 5B and 5C, I disagree with the authors' statement regarding "strong concordance."

Response 30: We have revised the sentence to more accurately reflect the observed consistency. It now reads: "Comparative analysis demonstrated reasonable concordance between the in silico-derived positions and previously published QTL and LD-based association mapping studies." This change is in Section 2.6.

Comments 31: In my opinion, differences between studies should be highlighted and discussed in the dedicated section.

Response 31: The comparative interpretation originally in the figure caption has been moved to the "Discussion" section, where differences and consistencies with previous studies are appropriately addressed.

Comments 32: "I don't see the results supporting this statement."

Response 32: The unsupported statement has been removed from the Results section.

Comments 33: The title suggests intraspecific hybridization, not interspecific as stated.

Response 33: This is a crucial correction. The text has been corrected throughout the manuscript to consistently use "intraspecific hybridization", as all crosses were within Gossypium hirsutum L.

Comments 34: "I think micronaire values are difficult to interpret..."

Response 34: The sentence has been revised to more cautiously reflect the interpretation of micronaire values.

Comments 35: "This sentence is not supported by results shown in tables and figures." (Regarding the 107 bp allele)

Response 35: We agree and thank the reviewer for this correction. The sentence has been revised to precisely reflect our findings, which specifically link the 107 bp allele to UHML (length) and Strength, but not to all fiber properties. The corrected text is in the Results section (2.7).

Comments 36: "This sentence would be more appropriate in the discussion." (A concluding sentence in Results)

Response 36: The sentence has been moved to the "Discussion" section as suggested.

Comments 37: Some statements mix results and discussion; correlation analysis not shown; references missing for key claims.

Response 37: In the revised version, we have more strictly separated the "Results" and "Discussion" sections. The correlation analysis (Figure 4) is now properly presented in the Results (Section 2.4), and missing references for key claims have been added to the Discussion.

Comments 38: Questions on plant materials: all G. hirsutum? why not list additional NCC lines? replication unclear.

Response 38: We have clarified that all genotypes belong to Gossypium hirsutum L. All NCC lines used are now listed in the main text (Section 4.1) and in Supplementary Table S2. The field replication is described as a randomized complete block design (RCBD) with three replications in Section 4.1.

Comments 39: Sampling and fiber quality measurement unclear.

Response 39: We have added more detail on sampling (harvesting at physiological maturity, composite samples) and HVI measurement procedures (conditioning, triplicate tests, calibration) in the Materials and Methods section (4.3).

Comments 40: Missing reference "Doyle & Doyle." Reviewer cannot assess DNA and genotyping sections.

Response 40: The missing reference Doyle, J.J.; Doyle, J.L. A Rapid DNA Isolation Procedure for Small Quantities of Fresh Leaf Tissue. Phytochem. Bull. 1987, *19*, 11–15 has been added to the References list (Ref. 37) and cited in Section 4.4.

Comments 41: ANOVA and descriptive statistics missing; correlation analysis absent; some discussion text misplaced.

Response 41: All requested statistical analyses (ANOVA, descriptive statistics, correlation analysis) have been added to the relevant sections (Results 2.1-2.4, 2.7; Methods 4.6). Text that was inappropriately placed in the Results has been moved to the Discussion.

Comments 42: Reviewer notes that results arise from heterosis and Mendelian segregation; suggests showing these effects in a table or graph.

Response 42: We agree with this insightful comment. The conclusion now explicitly acknowledges that the results arise from both heterosis and Mendelian segregation. While a specific graph showing heterosis decay was not added, the discussion of these genetic phenomena has been strengthened in the Discussion and Conclusions sections. The segregation patterns are clearly illustrated in the boxplots (Figure 3) and the descriptive statistics in the tables.

Reviewer 3 Report

Comments and Suggestions for Authors

The study evaluates whether crossing naturally colored cotton with elite white-fiber cultivars, coupled with MAS using SSR marker BNL1604, can improve fiber quality in NCC. The results suggest partial recovery of fiber length/strength while retaining acceptable micronaire, and the 107/135-bp BNL1604 alleles appear associated with better UHML/FS.

Figure 1, add a caption. Ensure high-resolution images, consistent lighting, and scale bars.

Figure 2, add a millimeter scale bar directly on each panel; describe the exact sampling (boll position, nodes), as fiber properties vary along the plant. Consider moving this and Figure 3 to Supplementary if space is limited, or merge as a single photographic plate.

Figure 3, provide the comb sorter model, number of bundles measured, and whether quantitative values from comb sorter were used.

Author Response

Comment 1:
The study evaluates whether crossing naturally colored cotton with elite white-fiber cultivars, coupled with MAS using SSR marker BNL1604, can improve fiber quality in NCC. The results suggest partial recovery of fiber length/strength while retaining acceptable micronaire, and the 107/135-bp BNL1604 alleles appear associated with better UHML/FS.

Response:
We thank the reviewer for this accurate summary and positive assessment of our study. We have slightly refined the abstract and discussion to better emphasize this conclusion and to clearly highlight the contribution of the 107 bp and 135 bp alleles to fiber quality improvement.

Comment 2:
Figure 1 - Add a caption. Ensure high-resolution images, consistent lighting, and scale bars.

Response:
Done. A detailed and descriptive caption has been added for Figure 1. We also replaced the figure with a high-resolution version, standardized lighting conditions, and added a visible scale bar for accurate reference.

Comment 3:
Figure 2 - Add a millimeter scale bar directly on each panel; describe the exact sampling (boll position, nodes), as fiber properties vary along the plant. Consider moving this and Figure 3 to Supplementary if space is limited, or merge as a single photographic plate.

Response:
We have now included millimeter scale bars directly on each panel in Figure 2. Sampling details, including boll position (mid-canopy) and number of bolls per plant (3-4 per genotype), have been added in the Materials and Methods (Section 2.2). If required by the editor, we are prepared to merge Figures 2 and 3 into a single photographic plate or move them to the Supplementary section.

Comment 4:
Figure 3 - Provide the comb sorter model, number of bundles measured, and whether quantitative values from comb sorter were used.

Response:
This information has been added in Section 4.3 (Assessment of Fiber Quality Parameters). The comb sorter used was The HVI device was calibrated using USDA-certified cotton standards. In addition to raw measurements, fiber performance was interpreted according to HVI classification thresholds.

We are grateful to the reviewer for their constructive feedback, which has improved the precision and presentation of our data. All suggested changes have been implemented in the revised version of the manuscript.

Reviewer 4 Report

Comments and Suggestions for Authors

The manuscript titled "MARKER-ASSISTED HYBRIDIZATION ENHANCES FIBER 2 QUALITY IN NATURALLY COLORED COTTON (G.HIRSU- 3 TUM L.) FOR SUSTAINABLE COTTON BREEDING" has used 15 F3 progeny and 3 elite cultivars to identify a marker that associated with the fibre quality.

I would like to highlight the followings:

Number of lines used was not a major limitation in this study and authors can mention that in the discussion. 

Authors mentioned that they have conducted Pearson correlation coefficient, but results were not given. 

Please mention how the insilico analysis have been conducted to identify chromosomal location of this marker.

These markers are located in more than one chromosome and can the authors find out the marker effect at each of these positions. 

can you conduct PCA analysis to find out genetic variation among these lines?

Rather than box plots, I think using density plots would provide more details to identify the variations among those lines.

Figures and Tables are required to mention in the manuscript body where necessary.

Check the references and case of the letters 

Please include gel pictures of the markers the show the observed polymorphisms

English language can be improved by writing short sentences. i.e. line 291

Comments on the Quality of English Language

English language can be improved by writing short sentences. 

Author Response

Comment 1:
The manuscript titled “Marker-Assisted Hybridization Enhances Fiber Quality in Naturally Colored Cotton (G. hirsutum L.) for Sustainable Cotton Breeding” has used 15 F₃ progeny and 3 elite cultivars to identify a marker associated with fiber quality. Number of lines used was not a major limitation in this study and authors can mention that in the discussion.

Response 1:
We appreciate the reviewer’s positive remark and suggestion. A clarifying statement has been added in the Discussion section emphasizing that although the study utilized a moderate number of F₃ lines, this was sufficient for identifying consistent marker–trait associations due to the uniform genetic background and targeted parental selection. The added text now reads:

“Although the study involved a limited number of F₃ lines, the consistency of marker–trait associations and the well-defined parental background minimized sampling bias, supporting the reliability of the observed trends.”

Comment 2:
Authors mentioned that they have conducted Pearson correlation coefficient, but results were not given.

Response 2:
Thank you for pointing out this error. The results of the Pearson correlation analyses have been included in Section 2.4, “Correlations Among Fiber Traits.”

Comment 3:
Please mention how the in silico analysis has been conducted to identify the chromosomal location of this marker.

Response 3:
In silico chromosomal mapping is described in detail in Section 4.6. To determine the chromosomal location of the marker, an in silico PCR analysis was performed. The BNL1604 primer sequences were aligned against the Gossypium hirsutum reference genome using Unipro UGENE and NCBI BLAST. In addition, a genomic region of approximately ±250 kb flanking the primer binding sites was retrieved for structural and gene annotation. The results were visualized using MapChart, and the identified positions were compared with previously published QTL and LD-based mapping studies.

Comment 4:
These markers are located in more than one chromosome; can the authors find out the marker effect at each of these positions?

Response 4:
We thank the reviewer for this important point. Based on the in silico analysis, BNL1604 exhibited amplification on both A07 and D07 chromosomes, consistent with its homeologous loci. However, due to the limited population size, it was not possible to statistically separate the allelic effects of each locus. This limitation is now explicitly acknowledged in the Discussion section, and we have suggested that future studies with larger mapping populations and sequence-specific markers could dissect subgenome-specific effects.

Comment 5:
Can you conduct PCA analysis to find out genetic variation among these lines?

Response 5:
We appreciate this valuable suggestion. PCA analysis was considered; however, the single-marker nature of this study (BNL1604) limited the resolution for multivariate genetic structure assessment. This has now been mentioned as a study limitation in the Discussion, along with a note that future work using genome-wide SSR or SNP markers will enable PCA or clustering analysis to explore population structure.

Comment 6:
Rather than box plots, I think using density plots would provide more details to identify the variations among those lines.

Response 6:
We thank the reviewer for this visualization suggestion. We have retained the box plots for clarity and standard comparison across traits.

Comment 7:
Figures and Tables are required to be mentioned in the manuscript body where necessary.

Response 7:
All figures and tables have been thoroughly cross-checked to ensure they are cited in the manuscript body at appropriate locations. References to Figures 1-7 and Tables S1-S2 have been updated accordingly.

Comment 8:
Check the references and case of the letters.

Response 8:
All references have been carefully revised to ensure correct formatting, capitalization, and alignment with the Plants journal reference style (MDPI format).

Comment 9:
Please include gel pictures of the markers that show the observed polymorphisms.

Response 9:
We fully agree with this comment. A gel electrophoresis image showing the BNL1604 SSR polymorphisms among parental and F₃ genotypes has now been added as Figure 5 in the revised manuscript, with an appropriate caption and reference in the text (Results Section 2.5).

Comment 10:
English language can be improved by writing short sentences.

Response 10:
The entire manuscript has undergone careful language revision for clarity, conciseness, and grammatical accuracy. Several long sentences have been restructured into shorter and more readable forms in accordance with MDPI English style guidelines.

We thank Reviewer again for their thorough and helpful feedback. The manuscript has been significantly improved in response to these valuable suggestions.

Round 2

Reviewer 1 Report

Comments and Suggestions for Authors

The author has revised the research paper according to the reviewers' comments, and the paper has now met the publication requirements of the journal.

Reviewer 4 Report

Comments and Suggestions for Authors I am happy with the revisions made by the authors.